# Molecular Mechanisms of Brassinosteroid-Mediated Responses to Changing Environments in *Arabidopsis*

**DOI:** 10.3390/ijms21082737

**Published:** 2020-04-15

**Authors:** Minghui Lv, Jia Li

**Affiliations:** Ministry of Education Key Laboratory of Cell Activities and Stress Adaptations, School of Life Sciences, Lanzhou University, Lanzhou 730000, Gansu, China; lvmh@lzu.edu.cn

**Keywords:** brassinosteroids, environments, light, temperature, water, stress, Arabidopsis

## Abstract

Plant adaptations to changing environments rely on integrating external stimuli into internal responses. Brassinosteroids (BRs), a group of growth-promoting phytohormones, have been reported to act as signal molecules mediating these processes. BRs are perceived by cell surface receptor complex including receptor BRI1 and coreceptor BAK1, which subsequently triggers a signaling cascade that leads to inhibition of BIN2 and activation of BES1/BZR1 transcription factors. BES1/BZR1 can directly regulate the expression of thousands of downstream responsive genes. Recent studies in the model plant Arabidopsis demonstrated that BR biosynthesis and signal transduction, especially the regulatory components BIN2 and BES1/BZR1, are finely tuned by various environmental cues. Here, we summarize these research updates and give a comprehensive review of how BR biosynthesis and signaling are modulated by changing environments and how these changes regulate plant adaptive growth or stress tolerance.

## 1. Introduction

Most higher plants are rooted in their habitats and cannot move around to select favorable growth conditions. They must deal with their living situations by constantly perceiving, processing, and responding to ever-changing environmental cues, such as light, temperature, water, nutrients, and microbes [1,2]. Many signaling molecules, such as phytohormones and their signaling cascades, are employed by plants to ensure their survival and optimal growth [3,4,5]. One such class of signaling molecules is brassinosteroids (BRs), a group of polyhydroxylated steroidal hormones widely distributed in the plant kingdom [6,7]. BRs play important roles in various biological processes during normal growth and development, as well as stress adaptation [8,9]. Mutants impaired in BR biosynthesis or signaling always show characteristic phenotypes, including severely retarded growth, male sterility, and altered responses to different stresses [6,10]. There are already many reviews summarizing BR biosynthesis, signal transduction, tissue-specific functions, and stress responses [11,12,13,14,15,16,17]. In this review, however, we mainly focus on how BR biosynthesis and signal transduction are finely tuned to coordinate growth and development with environmental changes.

## 2. Regulation of BR Biosynthesis, Catabolism, and Signal Transduction

To date, more than 70 BRs have been isolated in plants [18]. These BRs are mainly biosynthetic intermediates and catabolic products of known bioactive BRs, such as brassinolide (BL) and castasterone (CS) [8]. Like other plant steroids, BL is synthesized from a common sterol precursor termed cycloartenol [8]. After a series of chemical reactions including methylation, reduction, and desaturation, the BR-specific precursor campesterol (CR) is formed [19]. Steps from CR to BL are generally designated as the BR biosynthetic pathway [18]. Known enzymes catalyzing the BR biosynthetic pathway consist of a reductase DE-ETIOLATED2 (DET2) and several cytochrome P450s, including CONSTITUTIVE PHOTOMORPHOGENESIS AND DWARFISM (CPD), DWARF4 (DWF4), ROTUNDIFOLIA3 (ROT3), CYP90D1, and BRASSINOSTEROID-6-OXIDASES 1 and 2 (BR6ox1 and BR6ox2) [20,21,22,23,24,25,26,27,28,29,30,31,32]. Genes encoding these enzymes are transcriptionally regulated via a negative feedback loop to maintain BR homeostasis [33,34] (Figure 1). Furthermore, these genes can also be regulated by other factors (Figure 1). For instance, it was reported that *CPD* expression can be regulated by a putative transcriptional coregulator, BREVIS RADIX (BRX) [35]. It was also shown that CESTA, a basic helix-loop-helix (bHLH) transcription factor, acts as a positive regulator of *CPD* [36]. In addition, *DWF4*, whose encoding protein catalyzes a rate-limiting step in BR biosynthesis, is modulated by two bHLH transcription factors, TEOSINTE BRANCHED1/CYCLOIDEA/PROLIFERATING CELL FACTOR1 (TCP1) and PHYTOCHROME INTERACTING FACTOR 4 (PIF4) [37,38]. Both TCP1 and PIF4 are able to directly bind to the promoter regions of *DWF4* and positively regulate its expression [38,39]. Previous studies have demonstrated that the protein accumulation and transcriptional activity of PIF4 are regulated by temperature and light, thus connecting the external environmental changes with the internal BR biosynthesis [40,41,42,43]. Bioactive levels of BRs can also be determined by a number of catabolic modifications of BL and CS, such as C-26 hydroxylation catalyzed by a cytochrome P450 PHYB ACTIVATION-TAGGED SUPPRESSOR1 (BAS1), 23-O-glucosylation catalyzed by a UDP-glycosyltransferase UGT73C5, and a putative C-6 reduction step likely catalyzed by a dihydroflavonol 4-reductase (DFR)-like protein named BEN1 [44,45,46] (Figure 1). If the C-6 position ketone in BL is truly the target of BEN1, the resulting product should be an unstable molecule.

BL and its precursor CS are perceived by a cell-surface receptor kinase complex consisting of BRASSINOSTEROID INSENSITIVE 1 (BRI1) as the receptor and BRI1-ASSOCIATED KINASE 1 (BAK1) as the coreceptor, both of which belong to the leucine-rich repeat receptor-like kinase (LRR-RLK) protein family [47,48,49]. Both BRI1 and BAK1 have several functionally redundant paralogs [50,51,52]. BR perception results in enhanced physical interaction and reciprocal phosphorylation between the receptor and the coreceptor, leading to full activation of BRI1 [53,54,55,56,57,58]. Activated BRI1 then initiates a BR signal cascade by phosphorylating BR SIGNALING KINASE 1 (BSK1) and CDG1, two receptor-like cytoplasmic kinases (RLCKs) that are anchored to the cell membrane [59,60]. Activated CDG1 can subsequently phosphorylate and activate a protein phosphatase BSU1, which then dephosphorylates and inactivates a GSK3 kinase BRASSINOSTEROID INSENSITIVE 2 (BIN2) [60,61,62]. BIN2 is a negative regulator in the BR signaling pathway because it phosphorylates and destabilizes two well-characterized downstream transcription factors, BRI1-EMS-SUPPRESSOR 1 (BES1) and BRSSINAZOLE-RESISTANT 1 (BZR1), and presumably other four members of their entire subfamily [63,64,65,66,67]. BR-induced BIN2 inactivation allows the accumulation of nonphosphorylated BES1 and BZR1 in the nucleus, thereby promoting expression of thousands of their target genes [62,68]. The intensity of BR signaling can be controlled at multiple levels (Figure 2). At the receptor level, the function of BRI1 can be post-translationally regulated by PUB12/13-directed ubiquitination, PP2A-mediated dephosphorylation, BKI1-mediated kinase inhibition, and BIK1- and BIR3-mediated competition for the coreceptor BAK1 [69,70,71,72,73]. At the BIN2 level, it has been reported that BIN2 is regulated by OCTOPUS- or POLAR-mediated membrane sequestration, HDA6-mediated deacetylation, KIB1-mediated ubiquitination, TTL-enhanced interaction with BSU1, ABI1/2-mediated dephosphorylation, and ROS (reactive oxygen species)-mediated oxidation [74,75,76,77,78,79,80]. At the transcription level, it has been reported that PP2A phosphatases can promote BR signaling by dephosphorylating BES1 and BZR1, whereas 14-3-3 and BRZ-SENSITIVE-SHORT HYPOCOTYL1 (BSS1) negatively regulate BR signaling by inhibiting the translocation of BES1 and BZR1 from the cytosol to the nucleus [81,82,83]. Besides, protein degradation of BES1 and/or BZR1 has been reported to be regulated by many factors, including an F-box protein MORE AXILLARY GROWTH LOCUS2 (MAX2), ubiquitin E3 ligases such as PLANT U-BOX40 (PUB40), CONSTITUTIVE PHOTOMORPHOGENIC1 (COP1) and SINA of *Arabidopsis thaliana* (SINATs), photoreceptors such as UV RESISTANCE LOCUS 8 (UVR8), CRYPTOCHROME1 (CRY1) and PHYTOCHROME B (PHYB), and an autophagy receptor DOMINANT SUPPRESSOR OF KAR2 (DSK2) [84,85,86,87,88,89,90,91,92,93]. Moreover, BES1 and BZR1 were reported to be oxidized by ROS (e.g., H_2_O_2_) and reduced by a thioredoxin TRXh5 [94]. It was also shown that BES1 is phosphorylated by a mitogen-activated protein kinase 6 (MAPK6) to participate in immune response [95].

## 3. BR Biosynthesis, Signaling, and Function in Response to Various Environmental Conditions

### 3.1. Light

Light acts not only as an energy source, but also as an important environmental signal to regulate plant growth and development [96]. Light quality (spectral composition), quantity, direction, and duration (photoperiod) around a plant lead to its morphological and architectural fitness during photomorphogenesis, phototropism, and shade-avoidance responses [97]. Reports regarding light as signals were mainly about light with wavelength spectrum ranging from ~300 to ~750 nm, which are known to be perceived by several sets of photoreceptors including UVR8, cryptochromes, phototropins, and phytochromes. Phytochromes sense red (R) and far-red (FR) light (~600–750 nm) and mainly regulate photomorphogenesis, flowering transition, and shade-avoidance response [98]. Phytochromes exist as two photointerconvertible forms, the red light-absorbing form (Pr) that is inactive and the far-red light-absorbing form (Pfr) that is active. Pr is transformed to Pfr upon exposure to red light [99]. Conversely, Pfr is transformed to Pr upon exposing to far-red light [99]. The interconversion between Pr and Pfr impacts on the association of phytochromes with phytochrome interacting factors (PIFs), affecting their stability, and thus alters downstream gene expression profiles [99]. The cryptochromes and phototropins sense blue light (~400–500 nm) and mainly regulate circadian rhythms and photomovement responses such as phototropism, chloroplast movement, and stomatal opening [98]. UVB light (~300–315 nm) is perceived by a UVR8 homodimer, which is broken down into monomers upon UVB activation, the monomers subsequently interact with their targets to activate downstream signaling components [98]. 

Several lines of evidence suggest a crosstalk between light and BR signaling. Firstly, mutants defective in either BR biosynthesis or signaling exhibit constitutive photomorphogenic phenotypes, including shortened hypocotyls, opened and expanded cotyledons, when grown in darkness [20]. Secondly, an activation tagged BR-inactivating enzyme gene, *bas1-D*, suppresses the long hypocotyl phenotype caused by mutations in the photoreceptor phytochrome B (phyB) [44]. Thirdly, attenuation of light sensitivity by phyB mutation enhances BR sensitivity and partially suppresses the BR-deficient phenotypes of *bri1-5*, an intermediate mutant allele of *BRI1* [85]. 

Previous reports demonstrated that the expression of several BR biosynthetic genes, *CPD*, *DWF4*, and *BR6OX2*, could be induced in the dark and repressed in the light [100,101,102,103]. PIF4 is a transcription factor downstream of phytochrome-mediated light signaling [104]. The expression of *PIF4* is tightly controlled by the circadian clock, and the protein stability of PIF4 is regulated by light [105]. PIF4 can directly bind to the promoter regions of *CPD* and *DWF4* [38]. In *pifq*, a quadruple mutant of PIFs, the fluctuated expression patterns of these BR biosynthetic genes were completely blocked, suggesting that PIFs are essential to diurnal BR biosynthesis [102]. A recent study found that blue light perception in above-ground tissues is important for *DWF4* mRNA accumulation in root tips and therefore is important for root growth [106]. Another previous report showed that during seedling photomorphogenesis, light inhibits hypocotyl elongation by activating the expression of two BR-inactivating enzymes, *BAS1* (*CYP734A1*) and *SOB7* (*CYP72C1*) [107].

Light also affects BR signaling (Figure 3A). The transcription levels of *BES1* and *BZR1* are elevated at night and decreased during the day [102]. Interestingly, tracing BES1 protein level showed an inconsistency between protein accumulation and transcript levels, suggesting BES1 can be controlled post-translationally [102]. In fact, it was reported that the protein stability of BES1 and BZR1 is also regulated by light. COP1, a dark-activated ubiquitin ligase, is able to capture and degrade phosphorylated BZR1 in darkness [85]. A group of SINAT E3 ligases, on the contrary, are involved in degrading nonphosphorylated BES1 protein in the light [86]. In addition to light-controlled protein abundance of BES1 and BZR1, recent studies demonstrated that light negatively regulates the transcriptional activity of BES1 and BZR1 [88,89,90,92]. When illuminated by red and blue light, respectively, phytochrome B (phyB) and cryptochrome 1 (CRY1) interact specifically with nonphosphorylated BES1 and BZR1. This light-dependent interaction leads to the inhibition of BES1 and BZR1 DNA-binding activity and ultimately the expression of their target genes [88,89,90,92]. Moreover, blue light-activated CRY1 also interacts with BIN2 and thus enhances the interaction between BIN2 and BZR1. Consequently, BZR1 phosphorylation is enhanced and its nuclear accumulation is inhibited [90].

According to these studies, under normal light conditions, the photoreceptors are activated and associated with BES1 and BZR1, BR signaling is therefore repressed (Figure 2). Under a shaded or dark environment, however, the reduced light intensity leads to dissociation of photoreceptors from BES1 and BZR1 and hence release the repression of the BR signaling. The derepressed BR signaling, alone or together with other hormonal cascades, promotes growth so that the plants are able to get enough light source for better growth. For example, it was reported that BR signaling functions synergistically with the auxin pathway to stimulate hypocotyl and petiole elongation in response to shade [108]. Moreover, it was shown that BZR1 interacts with PIF4 in integrating the hormonal and light signals in response to diurnal growth [41] (Figure 2).

Temperature is an important environmental factor that governs the seasonal and diurnal behavior of plants grown in temperate geographical zones [109]. Many plants are highly responsive to even mild changes in temperature and adjust their growth and development accordingly [110]. Temperature elevation within the growth-permissive range leads to the elongation of hypocotyls and petioles in seedlings, hyponastic growth of petioles, and accelerated flowering in adults, which are collectively called thermomorphogenesis [111,112]. Temperatures beyond the growth-permissive range, however, result in cold- or heat-stress conditions that evoke stress responses instead of normal growth [110]. Recent studies showed that BR biosynthesis and signaling are affected by temperature and are involved in thermomorphogenesis and stress adaptations in response to temperature changes [113] (Figure 3B). 

In thermomorphogenesis, the photo receptor phytochrome B (phyB) was also identified as a major temperature sensor [114,115]. Besides the Pfr-to-Pr transformation by far-red light illumination, the Pfr can revert back to Pr via dark reversion [114,115]. A thermal signal accelerates Pfr-to-Pr dark reversion and facilitates the formation of phyB nuclear bodies, which inhibits phyB from interacting with and repressing PIF4 [115]. PIF4 is a transcription factor whose role has been established in both light and gibberellin (GA) signaling, as well as in thermomorphogenic growth [40,104,116]. Once stabilized by warm, PIF4 can promote the expression of a number of BR and auxin biosynthetic genes, such as *CPD*, *DWF4*, and *YUCCA5* [37,102,117]. In addition, thermal-caused BR accumulation via PIF4 triggers nucleus translocation of BZR1, which in turn associates with PIF4 to upregulate the expression of cell elongation-related target genes [118]. In contrast to the promotive effects of thermal signaling on BR signaling in aerial parts of plants, it seems that prolonged warm temperature attenuates BR signaling in roots [119]. A recent report demonstrated that BR-defective mutants are hypersensitive, whereas BR signaling-activated mutant *bes1-D* showed reduced sensitivity to warm-induced root growth acceleration [119]. The study also found that the BRI1 protein abundance was downregulated by prolonged warmer condition due to internalization and degradation [119].

Temperature below the growth-permissive range leads to cold stresses, including chilling stress (0–15 °C) and freezing stress (<0 °C) [120]. Plants usually display a basal-level freezing tolerance to low-temperature conditions [121]. This basal tolerance can be further enhanced by cold acclimation, a process in which plants acquire increased freezing tolerance when pre-exposed to low but nonlethal temperatures [121]. *COLD-REGULATED* (*COR*) genes play fundamental roles in plant tolerance to cold stress, many of which encode osmolytes and cryoprotective proteins that protect cells from freezing damage [122]. Some *COR* genes contain a conserved cis-element termed *C*-*REPEAT*/*DEHYDRATION-RESPONSIVE ELEMENT* (*CRT*/*DRE*), which is directly bound by the CRT/DRE BINDING FACTOR (CBF/DREB) family transcription factors [122]. The expression of these *COR* genes, therefore, is transcriptionally activated by CBFs [123]. There are also several *COR* genes that are cold-regulated but in a CBF-independent manner [124]. Expression of *CBFs* is tightly controlled by cold, with induction within 15–30 min, peaks at 1–3 h, and declines afterwards [125]. Several transcription factors were reported to positively or negatively regulate the expression of *CBFs* [126]. For example, INDUCER OF CBF EXPRESSION 1 (ICE1) is an MYC-type bHLH transcription factor responsible for the transcription of *CBFs* by directly binding to their promoters under cold stress [127]. Although *ICE1* expression is slightly upregulated by cold, its protein stability is predominantly controlled by posttranslational modifications including phosphorylation, ubiquitination, and SUMOylation, in response to cold stress [127,128,129].

BRs were shown to play a promotive role in plant resistance to the cold environment as they increase chilling tolerance when externally applied [130]. Moreover, BR-deficient mutants are hypersensitive, while BR signaling-activated mutants display increased tolerance, to freezing stress [131,132]. Recent studies in Arabidopsis uncovered the molecular mechanisms by which BR signaling contributes to basal and the acquired freezing tolerance through cold acclimation [129,132,133]. BIN2 has been reported to play a dual role during such cold stress response [129]. In the early time (<1 h) when the plants are exposed to cold, the kinase activity of BIN2 is decreased, leading to the accumulation of nonphosphorylated BZR1 and SUMOylated CESTA, another BIN2 substrate whose target specificity is altered by phosphorylation and phosphorylation-repressed SUMOylation [129,132,133]. Activated BZR1 and SUMOylated CESTA directly target to *CBFs* and hence induce the CBF-dependent *COR* gene expression [132,133]. Activated BZR1 also regulates the mRNA abundance of CBF-independent *COR* genes to modulate plant response to cold stress [133]. At a later time, however, BIN2 kinase activity recovers, leading to the inhibition of BZR1 and CESTA, as well as ICE1 [129]. BIN2 interacts with and phosphorylates ICE1, destabilizing ICE1 and attenuating its transcriptional activity [129]. BIN2-accelerated ICE1 degradation is partially responsible for the decline of *CBF* expression and was considered as a strategy for balancing plant growth and cold stress responses [129].

Heat stress is caused by high temperature (above the growth-permissive range), which affects photosynthesis, alters membrane fluidity, and disrupts the overall balance of metabolic processes [134]. Heat stimulates the excessive accumulation of ROS, leading to oxidative damages of lipids, proteins, and DNA [134]. Induction of heat tolerance is conferred by many processes in plants, including induction of heat shock proteins (HSPs), protection of membrane integrity, and recovery of protein activity [135]. In addition, plants also employ ROS-eliminating enzymes such as superoxide dismutase (SOD), ascorbate peroxidase (APX), glutathione reductase (GR), peroxiredoxin (Prx), and catalase (CAT) to protect plants from heat stress [135]. In contrast to our knowledge of BR-involved thermoadaptive growth and cold tolerance, we know little about BR-mediated heat stress response. Recent studies demonstrated that exogenous application of BRs can enhance heat tolerance by facilitating photosynthesis, keeping membrane integrity, and maintaining proper redox status [6,136,137].

### 3.2. Water Availability

Water is vital for plants due to its irreplaceable roles in various physiological processes. Water in soil, which is greatly affected by climate change, is an important constraint for plant growth and distribution [138]. Plants absorb water mainly through their root systems, whose growth orientation and architecture, in turn, are adaptively determined by water distribution in the soil (or soil moisture gradients). This plastic root growth in response to soil moisture gradient is called hydrotropism [139]. A recent study using different Arabidopsis ecotypes suggested a possible role of BRs in hydrotropism [140,141] (Figure 3C). Such a scenario is supported by the observation that the strong hydrotropic response in Wassileskija (Ws) ecotype is reduced by either mutation of the BR receptor BRI1 (*bri1-5*) or treatment with a BR biosynthetic inhibitor, BRZ [140]. The study also demonstrated a direct interaction between BRI1 and a plasma membrane H^+^-ATPase AHA2. The authors proposed that this may contribute to the strong hydrotropism observed in Ws [140,141]. Whether moisture gradients affect BR asymmetric distribution in root tips, however, was not revealed.

Drought is one of the most common abiotic stresses against plant growth and even survival [142]. Several studies revealed roles of BRs in plant tolerance to drought, acting mainly in crosstalk with abscisic acid (ABA) signaling [79,143,144,145,146]. The proposed ABA signaling pathway consists of a group of ABA receptors named PYRABACTIN RESISTANCE1/PYR1- LIKE/REGULATORY COMPONENTS OF ABA RECEPTORS (PYR/PYL/RCAR), a family of type 2C PROTEIN PHOSPHATASES (PP2Cs, e.g., ABI1 and ABI2) as the coreceptors negatively regulating ABA signaling, a class of Snf1-Related Kinase 2s (SnRK2s) as positive regulators, and a class of ABA-responsive element-binding factors (ABFs) as the downstream transcription factors [147]. ABA induces the interaction between receptors PYR/PYL/RCAR and coreceptors PP2Cs, leading to the degradation of PP2Cs and thus releases SnRK2s from PP2C-mediated dephosphorylation and inhibition [147]. The freed SnRK2s then interact with and phosphorylate ABFs to regulate ABA-responsive gene expression [147]. ABA and BR act antagonistically to balance growth and stress response. Two pathways are converged at BIN2 whose activity is repressed by PP2Cs through dephosphorylation [79]. During drought response, increased ABA production leads to the repression of PP2Cs and the enhanced activity of BIN2 [79,147]. BIN2 restricts plant growth by attenuating BES1/BZR1 activity, but enhances drought tolerance by phosphorylating SnRK2s and intensifying ABA signaling [144,146]. In addition, BIN2-mediated trade-offs between growth and drought tolerance implicate a group of transcription factors including WRKY46, WRKY54, and WRKY70 [148]. During normal growth, these transcription factors interact with BES1 to promote the transcription of BR target genes while repressing drought-responsive genes [148]. Under drought conditions, on the other hand, BIN2 phosphorylates and destabilizes WRKY54 to reduce BR-stimulated growth but allow expression of drought-inducible genes [148]. Different from the WRKYs, TINY is a transcription factor that is stabilized via BIN2 phosphorylation under drought conditions [149]. TINY positively regulates drought responses by activating drought-responsive genes while inhibits BR-mediated growth through TINY-BES1 antagonistic interactions [149]. RD26 is a NAC transcription factor induced by drought and ABA, and functions to increase plant drought tolerance through promoting drought-responsive gene expression [150]. *RD26* is one of the target genes of BES1 and is repressed by BR signaling [150]. Drought-induced BR signaling blockage releases RD26 from transcriptional inhibition and allows accumulation of RD26 protein, which further restricts the BR signaling by binding to BES1 and antagonizing its transcriptional activity [150].

### 3.3. Minerals and Ions

Mineral nutrients, mainly taken up from the soil, are essential for normal plant growth and development [151]. Important progress has been made in understanding the interplay between ion and BR biosynthesis or signaling. Calcium is a universal messenger involved in the regulation of various developmental and adaptive processes in response to physiological and environmental stimuli [152]. A Ca^2+^ signal can be sensed by diverse Ca^2+^ binding proteins such as calmodulins (CaM) [152]. A previous study showed that Ca^2+^/CaM promotes BR biosynthesis through interacting with and activating DWARF1, a Ca^2+^-dependent calmodulin-binding protein responsible for converting 24-methylenecholesterol to CR, the initial BR biosynthetic precursor [153]. Perception of BRs by BRI1, in turn, has been reported to cause an elevation in cytosolic Ca^2+^, thus initiating a Ca^2+^ signaling cascade in the cytosol [154]. Further evidence suggests that a BR-dependent elevation in cyclic GMP (cGMP) may be involved in the BR-initiated cytosolic Ca^2+^ signaling [154]. This notion is supported by another recent study which revealed that BRI1 possesses a cGMP-generative ability that is conferred by a guanylate cyclase catalytic center encapsulated in the BRI1 kinase domain [155]. Similar scenarios regarding the interplay between ion and BRs have also been seen in iron ions. A recent report found that low iron availability in soil accelerates Arabidopsis root growth by activating BR signaling, whereas high iron concentration shows an opposite effect [156]. Altered hormone signaling intensity, in turn, modulates iron accumulation in the root elongation and differentiation zones, constituting a regulatory feedback loop between BR and iron [156]. The BR receptor inhibitor BKI1, the transcription factors BES1/BZR1, and the ferroxidase LPR1 act as core operators in this feedback loop [156]. The report revealed that this feedback regulatory machinery between iron and BR signal transduction also plays a role in adaptive root growth in response to soil phosphate availability [156]. 

Except for its involvement in coordinating plastic growth in response to mineral nutrients, it was demonstrated that BRs help plants to tolerate various ion stresses, such as salinity [6]. Exogenous BR application increases germination rate, promotes growth and productivity of a number of crops under salty conditions [157,158]. Several possible mechanisms by which BRs alleviate salt stress or ion toxicity in plants were proposed [159]. For example, BR signaling might be involved in controlling the water loss under high salinity conditions via regulating stomata density and stomatal conductance [160,161]. BRs might regulate plasma membrane localized cation and anion channels that implicate in ion stress response [162,163]. It was not until recently that two studies proposed a molecular mechanism by which BR signaling dynamically respond to salt stress and positively contribute to salt tolerance [164,165]. In the early stage of salt stress response, salinity stimulates the protein accumulation of ULP1a, a deSUMOylating enzyme which is able to deSUMOylate BZR1 and facilitate its phosphorylation by BIN2 and ultimately degradation, thus inhibiting BR signaling [165]. This temporally suppressed BR signaling, however, recovers soon due to BR-induced degradation of ULP1a [165]. ULP1a was reported to inhibit growth during salt stress and *ulp1a* mutants were salt-tolerant [165]. Taken together, the antagonistic regulation of ULP1a by salinity and BRs seems to be a molecular switch determining plant growth and salt stress response. 

### 3.4. Microbes and Immune Response

Microbial pathogens constitute an important aspect of biotic stress challenging plant growth and development. To effectively combat the invasion of pathogens, multiple layers of defense mechanisms have been developed by plants during evolution, including preformed structural barriers and inducible defense mechanisms [166]. Once pathogens pass the structural barriers, they are likely to be recognized by plants with specific pattern recognition receptors (PRRs) via sensing pathogen-associated molecular patterns (PAMPs) [167]. Perception of PAMPs by PRRs activates PAMP-triggered immunity (PTI), the first layer of the inducible defense responses which is believed to fend off most microbes [167]. The representative PTI response in Arabidopsis is mediated by FLAGELLIN SENSITIVE 2 (FLS2), an LRR-RLK that recognize the bacterial flagellin (or its active epitope flg22) [168,169]. flg22-perception induces heterodimerization and transphosphorylation between FLS2 and BAK1, leading to activation and dissociation of BSK1 and BOTRYTIS-INDUCED KINASE1 (BIK1) from FLS2 [170,171,172,173]. While BIK1 phosphorylates NADPH oxidase to trigger a rapid burst of ROS, BSK1 phosphorylates MAPK Kinase Kinase 5 (MAPKKK5) to activate a MAPK cascade, ultimately causing flg22-induced transcriptional changes [174,175,176].

BRs are increasingly reported to be involved in plant response to pathogen attack (Figure 3D). The roles of BRs in plant defense, however, are controversial from different studies [177,178]. Some studies suggested that BRs inhibit flg22-induced PTI response, evidenced by, (1) BR treatment significantly decreased flg22-induced PTI responses, including oxidative burst and defense gene expression; (2) flg22-induced MAPK activation was enhanced in BR-defective mutants such as *bri1* and *bin2-1* [179,180]. Other studies, however, proposed a positive role of BRs on plant immunity, as supported by, (1) Expression of defense response genes, such as *PR1*, *PR2*, and *PR5*, is remarkably lower in a BR biosynthetic gene mutant *cpd* than that in wild type; (2) Exogenous application of BRs conferred diverse crops resistance to a broad array of pathogens [22,178]. Alternatively, the exact effect of BRs in plant pathogen response seems to vary in different plant species, different developmental stages, as well as diversified pathogen types used for test [181]. In a reverse aspect, it was reported that although flg22-induced activation of PTI does not affect BR signal transduction, it inhibits the expression of many BR biosynthetic genes [182]. Interestingly, this inhibitory effect did not require BR perception or signaling, and occurred within 15 min of flg22 treatment [182]. The flg22-induced transcriptional inhibition of BR biosynthetic genes may result in reduction in BR content and may contribute to a negative and bidirectional crosstalk between PTI and BRs.

Since BRI1-mediated BR signaling and FLS2-mediated PTI response share BAK1, BIK1, and BSK1 as their common regulatory components, it is logical to speculate that these proteins may contribute to the crosstalk between BR signaling and PTI response [72,170,171]. However, it was shown that BAK1 function in cell-death control and innate immunity is independent of its function in BR signaling and that BR-controlled inhibition of PTI response takes place downstream and independently of BAK1 [179,183]. This notion was supported by a recent study which identified several conserved BAK1 phosphorylation sites that are specifically required for its immune function but not for BR signaling regulation [184]. BIK1 is a receptor-like cytoplasmic kinase that negatively regulates BR signaling while positively regulating the FLS2-mediated PTI response [72]. Its functions in both processes, however, were mechanistically uncoupled [72]. BSK1 positively regulates both BR and PTI signaling, but its functional importance in two different pathways seems to be different [59,171,172]. Inhibition of BSK1 increased susceptibility to pathogens, but had no obvious effect on BR response [59,171]. These findings raise the possibility that competition for BSK1 between BRI1 and FLS2 may contribute to BR-mediated PTI inhibition.

MAPK cascades, consisting of MAPK kinase kinases (MAPKKKs), MAPK kinases (MKKs), and MAPKs, play important roles in diverse biological processes [185]. For example, MKK4/MKK5-MPK3/MPK6 was reported to be activated not only by MAPKKK3/5 to participate in flg22-induced immune response, but also by MAPKKK YODA (also known as YDA or MAPKKK4) to regulate stomata development [180,186]. These two regulatory pathways, interestingly, were antagonistic because of competing for interaction with MKK4/5 by MAPKKK3/5 and YDA [180]. Recent reports demonstrated that BIN2 interacts with and phosphorylates YDA to inhibit its activity, raising the possibility that BR may inhibit FLS2-mediated PTI response through inhibiting BIN2 activity and activating a YDA-dependent MAPK cascade [160] (Figure 3D).

Despite these aforementioned studies suggesting an interaction between BR signaling and PTI response at the level of the receptor or downstream cytoplasmic kinases, it was speculated that this interaction might only make a minor contribution to the suppression of PTI [187]. The trade-off between BR-mediated plant growth and immune response mainly occur at the transcriptional level (Figure 3D). Recent studies revealed a sufficient role of BZR1 in repressing PTI response [188]. Upon activation by BR, BZR1 promotes the expression of WRKY transcription factors (such as WRKY11, WRKY15, and WRKY70) to negatively regulate plant immunity [188]. In addition, BR-activated BZR1 also promotes the expression of transcription factor HOMOLOG OF BRASSINOSTEROID ENHANCED EXPRESSION2 INTERACTING WITH IBH1 (HBI1), a negative regulator of immunity that was shown to repress a subset of genes involved in immunity and PTI-induced growth arrest [189,190]. Taken together, these studies establish a mechanism by which BZR1 functions as an integrator to balance plant growth and immune response.

## 4. Conclusions and Future Perspectives

BRs have been reported to function in a wide spatial-temporal territory during the entire life span of plants [191]. However, the available evidence indicates that endogenous BRs do not undergo long-distance transport [192]. Therefore, control of BR biosynthesis, catabolism, or signaling at the tissue or organ level is crucial to optimize plant growth in response to developmental stages and environmental changes. Although many internal factors have been identified to mediate BR biosynthesis and signaling, how these factors integrate with certain developmental stages or multiple environmental factors remains largely unknown [191]. Understanding the precise regulatory mechanisms of BR biosynthesis and signaling will be beneficial for future crop improvement via traditional breeding or genetic engineering such as CRISPR/Cas-based genome editing.

## Figures and Tables

**Figure 1 ijms-21-02737-f001:**
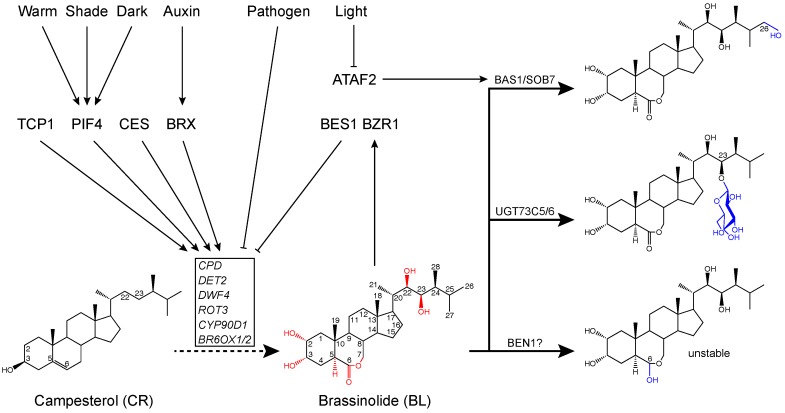
Simplified model of brassinosteroid (BR) homeostasis maintained by biosynthesis, catabolism, and their regulatory networks. Brassinolide (BL), the most active BR, is biosynthesized from the BR-specific precursor campesterol (CR) through a number of catalytic steps. BL can be inactivated via several modifications including hydroxylation, glucosylation, and reduction. BR biosynthetic and metabolic genes are transcriptionally regulated by several internal factors in response to environmental cues. Arrows indicate stimulation, whereas lines with blunt ends represent suppression.

**Figure 2 ijms-21-02737-f002:**
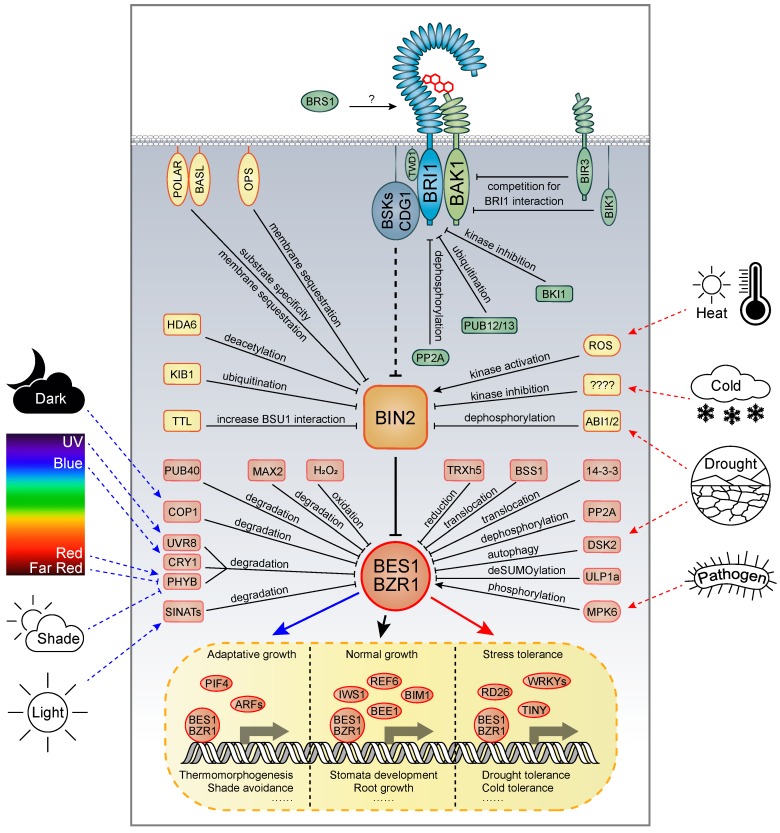
An integrated model for BR signal transduction and multiple levels of regulatory mechanisms in response to environmental changes. At the receptor level, the function of BRI1 is negatively regulated by PUB12/13, PP2A, BKI1, BIK1, and BIR3, and positively regulated by BRS1, TWD1, and BAK1. At the BIN2 level, the function of BIN2 is inhibited by OCTOPUS, POLAR, HDA6, KIB1, TTL, ABI1/2, and ROS. At the transcription level, the protein stability of BES1 and BZR1 is negatively regulated by MAX2, PUB40, COP1, SINATs, PHYB, CRY1, and UVR8. BES1 or BZR1 also acts synergistically of antagonistically with many other transcription factors to determine plant growth and stress tolerance. Blue arrows indicate adaptive growth, whereas red arrows represent stress tolerance. Lines with blunt ends represent suppression.

**Figure 3 ijms-21-02737-f003:**
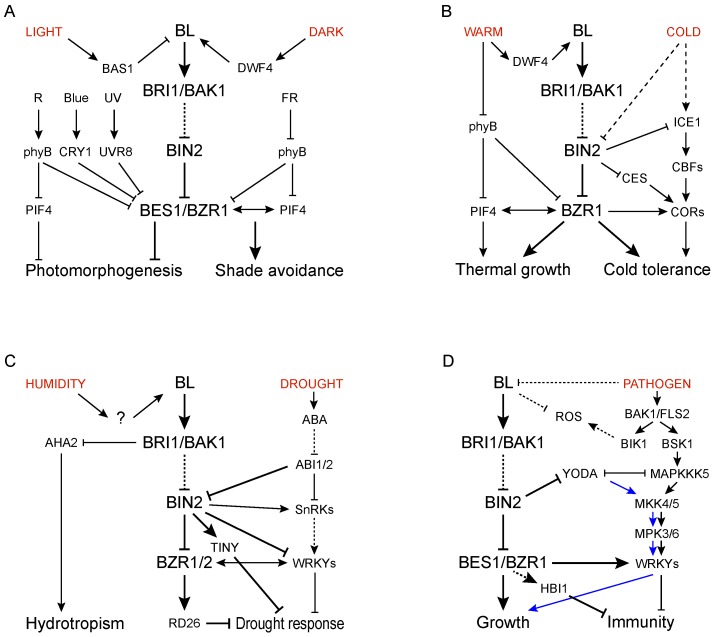
Schematic models indicating the contributions of BR biosynthesis and signaling in response to (**A**) light, (**B**) temperature, (**C**) water availability, and (**D**) pathogens. Arrows indicate activation, whereas lines with blunt ends represent suppression.

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
