# Peer review of "Molecular Mechanisms of Brassinosteroid-Mediated Responses to Changing Environments in Arabidopsis"

_ijms, 2020, doi:10.3390/ijms21082737_

Round 1

Reviewer 1 Report

I attached revision as pdf.

Author Response

Article is devoted to review molecular mechanisms of BR action in plants. I cannot find/see any figures in article although they are cited in text.
Our response: This might be a mistake during our previous submission. We uploaded two separate docx files, one is the main text, and the other contains the figures and figure legends. In the revised version, we have merged the figures and figure legends with the text file.
I have only some instructions:
1. TITLE. Article is strongly focused on molecular mechanisms of BR action in plants, generally physiological part is neglected. I do not say it is wrong but this is the reason that title should be changed. Much better would be for example: “Insight into molecular mechanisms of BR-mediated plant response to changing environmental conditions”.
Our response: Thanks for your suggestion, we have modified the title in the revised version. We would like to keep the title short, therefore we omit “Insight into” from the title you suggested. Because all the papers cited were about Arabidopsis, we put “Arabidopsis” in the title.
2. ABSTRACT. Abstract is little poor. Since work is devoted mainly to molecular mechanisms it should be somehow mentioned in abstract.
Our response: Thanks, in the revised version, we rewrote the abstract to make it more precise
3. CHAPTER 2: AN OVERVIEW OF BRASSINOSTEROID HOMEOSTASIS, SIGNAL TRANSDUCTION, AND REGULATION.
In my opinion better would be: “Regulation of BR biosynthesis and BR signal transduction”. Perhaps two chapters should be made: “Regulation of BR biosynthesis” and “Genetic background of BR signal transduction”.
Our response: Thanks for the suggestion, we have changed the subtitle in the revised version. We prefer the first recommended subtitle.
4. Chapter 3. BR Homeostasis, Signaling and Function in Various Environmental Conditions. Authors use here very general word ‘BR homeostasis” – is this justified? Perhaps ‘BR biosynthesis’ would be better.
Our response: Thanks for the advice, we have modified this subtitle in the revised version.
5. In chapter 3.1. Light and BRs – would be good to cite:
Asahina M., Tamaki Y., Sakamoto T., Shibata K., Nomura T., Yokota T. 2014. Blue light promoted rice leaf bending and unrolling are due to up-regulated brassinosteroid biosynthesis genes accompanied by accumulation of castasterone. Phytochemistry 104: 21-29.
Our response: Thanks for the advice. We cited this article as ref. [103] in the revised version.
6. In charter 3.2 “Temperature and BRs” authors wrote: “In contrast to our knowledge of BR involved thermo-adaptive growth and cold tolerance, we know little about BR-mediated heat stress response. Recent studies demonstrated that exogenous application of BRs can enhance heat tolerance by facilitating photosynthesis, keeping membrane integrity and maintaining proper redox status [6, 135]. The underline molecular mechanisms, however, are yet to be elucidated.”
I suggest checking the following articles:
Dhaubhadel, S., Browning, K. S., Gallie, D. R., Krishna, P.: Brassinosteroid functions to protect the translational machinery and heat-shock protein synthesis following thermal stress. The Plant Journal. 29: 681-691, 2002.
Kagale, S.; Divi, U.K.; Krochko, J.E.; Keller, W.A.; Krishna, P. Brassinosteroid confers tolerance in Arabidopsis thaliana and Brassica napus to a range of abiotic stresses. Planta 2007, 225, 353–364.
Sadura I., Libik-Konieczny M., Jurczyk B., Gruszka D., Janeczko A. 2020. Accumulation of H+-ATPase and the aquaporin HvPIP transcript and protein in barley brassinosteroid mutants and their wild type cultivars growing at various temperatures. Journal of Plant Physiology. 244:153090.
Sadura I. Libik-Konieczny M. Jurczyk B. Gruszka D. Janeczko A. 2020. HSP transcript and protein accumulation in brassinosteroid barley mutants acclimated to low and high temperatures. International Journal of Molecular Sciences 21: 1889.
Our response: Thanks for the advice, we checked these articles and cited the 2nd article as ref. [] in the revised version. We did notice that there are a lot of research advances in other plant species such as barley, maize, and rice, but constrained by the huge information and limited space, we decided not to incorporate these findings into this review. Another reason we didn’t cite all the papers above is that this review is focused on discussing research updates in Arabidopsis (We already added “Arabidopsis” in the title).
7. Chapter: 3.5 Microbes, immune response, and BRs Interestingly some BR mutants may be resistant to pathogens. Perhaps it is worthy to mention the following findings:
Ali SS, Gunupuru LR, Kumar GB, Khan M, Scofield S, Nicholson P, Doohan FM. 2014. Plant disease resistance is augmented in uzu barley lines modified in the brassinosteroid receptor BRI1. BMC Plant Biol., 14:227.
Janeczko A. Saja D. Dziurka M. Gullner G. KornaÅ› A. Skoczowski A. Gruszka D. Barna B. 2019. Brassinosteroid deficiency caused by the mutation of the HvDWARF gene influences the reactions of barley to powdery mildew. Physiological and Molecular Plant Pathology 108: 101438
Our response: Thanks. It’s true that BR function in barley and other plant species to optimize their growth in various environmental conditions. But this review is intended to summarize the molecular mechanisms in Arabidopsis. To keep it simple, we decided not to mention studies in other plant species.
8. Authors often use rather ‘general’ word “BR homeostasis’ – would be good to explain what exactly they understand by it.
Our response: We have changed the word “BR homeostasis” in most places to “BR biosynthesis” or “BR catabolism”.

Reviewer 2 Report

The submitted manuscript by Minghui Lv and Jia Li reviews that the role of brassinosteroids in plant growth, development and stress adaptations.

It is well organized review which covers the current BR research topics regarding abiotic/biotic responses.

Since BR signaling is involved in many plant responses and it is very complicate, the information in this review would be helpful for further research works especially light and temperature.

However I couldn't find the figure in the submitted PDF, so I cannot mention the quality of illustration at this stage. It would be nice if the author upload it with the figure for the next version.

Nevertheless, for the main text part, I can recommend it to publish as a review article.

Reviewer 3 Report

The manuscript is a comprehensive review on one of the most interesting and intriguing issues of plant studies, hormonal regulation of the adaptive status of plant. The review summarize different aspects of biochemical response to light, temperature, water availability, mineral, ions and immune response.

I think the manuscript could be improved:

1) in the subtitles as "Microbes, immune response, and BRs", "Minerals, ions, and BRs" "BRs" can be easily removed since all the review is about BRs. Perhaps, it could be changed to  ""Microbes and immune response", ""Minerals and ions"

2) I suggest that the author add a generalized scheme that would summarize the main effects of the BRs and the directions of its actions in the considered aspects.

The manuscript can be accepted after these minor revisions. 

Kind regards

Author Response

The manuscript is a comprehensive review on one of the most interesting and intriguing issues of plant studies, hormonal regulation of the adaptive status of plant. The review summarizes different aspects of biochemical response to light, temperature, water availability, mineral, ions and immune response.
I think the manuscript could be improved:
1) in the subtitles as "Microbes, immune response, and BRs", "Minerals, ions, and BRs" "BRs" can be easily removed since all the review is about BRs. Perhaps, it could be changed to ""Microbes and immune response", ""Minerals and ions"
Our response: Thanks for the advice, we changed these subtitles in the revised version.
2) I suggest that the author add a generalized scheme that would summarize the main effects of the BRs and the directions of its actions in the considered aspects.
Our response: Thanks, this is actually a mistake in the previous manuscript submission since we uploaded two docx files, one is the main text, and the other contains figures and figure legends. In the revised version, we merged the figures and figure legends with the text file. Now you should be able see the figures.